# Finding the Needle in the Haystack: Serological and Urinary Biomarkers in Behçet’s Disease: A Systematic Review

**DOI:** 10.3390/ijms24033041

**Published:** 2023-02-03

**Authors:** Marta Arbrile, Massimo Radin, Davide Medica, Paolo Miraglia, Letizia Rilat, Irene Cecchi, Silvia Grazietta Foddai, Alice Barinotti, Elisa Menegatti, Dario Roccatello, Savino Sciascia

**Affiliations:** 1Department of Clinical and Biological Sciences, School of Specialization of Clinical Pathology, University of Turin, 10124 Turin, Italy; 2Center of Excellence on Nephrologic, Rheumatologic and Rare Diseases (ERK-Net, ERN-Reconnect and RITA-ERN Member) with Nephrology and Dialysis Unit, San Giovanni Bosco Hub Hospital, University of Turin, 10124 Turin, Italy

**Keywords:** Behçet’s disease, biomarkers, diagnosis, disease activity, autoinflammatory disease

## Abstract

Urinary and serological markers play an essential role in the diagnostic process of autoimmune diseases. However, to date, specific and reliable biomarkers for diagnosing Behçet’s disease (BD) are still lacking, negatively affecting the management of these patients. To analyze the currently available literature on serological and urinary BD biomarkers investigated in the last 25 years, we performed a systematic literature review using the Population, Intervention, Comparison, and Outcomes (PICO) strategy. One hundred eleven studies met the eligibility criteria (6301 BD patients, 5163 controls). Most of them were retrospective, while five (5%) were prospective. One hundred ten studies (99%) investigated serological biomarkers and only two (2%) focused on urinary biomarkers. One hundred three studies (93%) explored the diagnostic potential of the biomolecules, whereas sixty-two (56%) tested their effect on disease activity monitoring. Most articles reported an increase in inflammatory markers and pro-oxidant molecules, with a decrease in antioxidants. Promising results have been shown by the omics sciences, offering a more holistic approach. Despite the vast number of investigated markers, existing evidence indicates a persistent gap in BD diagnostic/prognostic indices. While new steps have been taken in the direction of pathogenesis and disease monitoring, international efforts for the search of a diagnostic marker for BD are still needed.

## 1. Introduction

Behçet’s disease (BD) is a multisystemic inflammatory condition often described as a part of the vasculitic spectrum, whose etiology, although not fully characterized, is attributed to a complex inter-relationship between the genetic background and the dysregulation of both the innate and the adaptive immune system [1]. Females and males are equally affected, with a worse disease progression in males due to ocular, vascular, and neurological involvement [2]. Diagnosis onset is collocated between 25 and 30 years old, although countries with a low disease prevalence may show a delayed time of diagnosis [3].

The distribution of BD is widespread; however, it is more prevalent in countries along the ancient “Silk Road”, from the Mediterranean area to the far east, where it is associated with the distribution of the major histocompatibility complex antigen HLA-B51 [4].Even though the first description of BD dates back to 1937, its diagnosis still relies entirely on clinical criteria [5], and a laboratory test to help identify patients with BD is still lacking. Unfortunately, the most common symptoms of BD, including oro-genital aphthae, skin lesions, arthritis, and uveitis, overlap with other autoimmune diseases, such as inflammatory bowel conditions or connective tissue diseases, and the differential diagnosis may become a real challenge [6,7]. Moreover, the disease course tends to be considerably prolonged, and it may take months or years before all the typical signs and symptoms appear. Unfortunately, for most patients, an early diagnosis of BD can be an unrealistic goal and having one or more biomarkers of BD could drastically change how BD is diagnosed and ultimately help clinical evaluation. A biomarker is a measurable characteristic of the body that may indicate a particular biological state or condition [8]. Biomarkers are employed in many fields of medicine, such as disease diagnosis, disease activity evaluation, prognosis, and therapy monitoring. Since the 1950s, new biomarkers for BD have been studied to be applicable to all populations in which the disease is prevalent. Still, there is no consensus on a shared biomarker for BD to be evaluated by testing in the diagnostic routine. Recently, omics sciences have helped solve this diagnostic gap in multiple diseases using a promising holistic approach, but at the moment, they are not yet integrated into standard clinical care [9,10]. A system based on omics sciences is also needed in BD—on the one hand, for diagnosing BD patients early; and, on the other hand, for identifying different profiles of BD patients based on their disease activity, prognosis, and response to therapy. In order to contribute to this growing field of research, this study aimed to systematically review the currently available literature on the identification and characterization of the clinical utility of serological and urinary BD biomarkers investigated in the last 25 years.

## 2. Methods

### 2.1. Literature Search Strategy

A detailed literature search screening Ovid MEDLINE, In-Process and Other Non-Indexed Citation, the National Library of Medicine’s (NLM), and the in-process database for Ovid MEDLINE, from inception to November 2021, was performed a priori to identify original articles analyzing the diagnostic role of urinary and serological biomarkers in BD. The Population, Intervention, Comparison, and Outcomes strategy (PICO) was adopted to identify the best keywords to use in database queries. The following keywords and medical subject heading (MESH) terms were used in all possible combinations using Boolean operators: Behçet’s syndrome; retinal vasculitis; biomarkers; inflammation mediators; immune checkpoint proteins; pathogen-associated molecular pattern molecules.

### 2.2. Selection of the Studies

We screened and selected full-text articles, analyzing the titles and abstracts. After the first screening phase, we evaluated the selected abstracts and the full texts to determine eligibility. Papers retrieved by the literature search but reporting insufficient data according to the chosen PICO strategy were excluded. The online search was limited to case-control, cohort, and case-series studies. Studies with a small sample size (n < 20), conference abstracts, reviews, and animal studies were excluded. Articles written in languages other than English were excluded. The selection and inclusion criteria were determined a priori.

We considered studies eligible if they met the following inclusion criteria:

Studies that included at least 20 patients diagnosed with BD following the current International Study Group Criteria [5];

Studies that analyzed urinary biomarkers, serological biomarkers, or both;

Ex vivo studies (in vitro studies were excluded).

Four independent reviewers (MA, DM, PM, and LR) systematically analyzed the abstracts and full texts of the articles meeting the inclusion criteria; any disagreements were resolved by consensus. If consensus could not be achieved, a third party (MR) provided an assessment of eligibility. As the data on eligibility were dichotomous (eligible: yes/no), agreement at both the title and abstract review and the full article review stages was determined by the calculation of Cohen’s kappa coefficient (k > 8). We performed the present study according to the PRISMA guidelines [11].

### 2.3. Data Extraction and Data Synthesis

Data were extracted in an electronic database, summarized, analyzed, and discussed. For each study, the following data were identified: study design, country of origin, type of biomarker, methods used for detection, sample size, pathergy and HLAB51 positivity, type of involvement (systemic/organ-specific), different marker concentrations in BD, and controls and measures of association. The homogeneity of studies was assessed per each diagnostic maker. Quantitative synthesis was considered inappropriate due to the heterogeneity among studies in the population set, the type of biomarker analyzed, and the methods used for the identifications used in different studies. Therefore, a qualitative narrative synthesis was performed.

## 3. Results and Discussion

### 3.1. Systematic Literature Search

We retrieved 637 articles from the initial search (Figure 1).

Three hundred forty-four studies were excluded after the title and abstract screening because they did not fit the selection criteria described above. We further assessed twohundred ninety-three studies for eligibility. We excluded one hundred eighty-two studies because they did not meet the inclusion criteria, were not focused on biomarkers, did not reach statistically significant results, or were not in English. Finally, one hundred eleven articles were eligible for the qualitative synthesis.

Figure 2 shows the number of studies per year included in this systematic review. Furthermore, Table 1 displays the main characteristics of the analyzed studies, including the number of patients, study design, biomarkers tested, and accuracy.

A total of 6301 patients with BD (1813 with active, 1543 with inactive BD, and 2945 cases in which the activity of BD was not addressed in the study) met the inclusion criteria and were further analyzed. There were 5163 included controls, consisting of 4171 healthy controls (HC) and 992 patients with autoimmune diseases (such as SLE, AR, SS, multiple sclerosis, and vasculitis). Most studies were retrospective, whereas six had a prospective design.

Considering the extensive geographical diffusion of BD, we analyzed the countries of origin in which all included studies was performed. The global map of Figure 3 shows the publication rate of the analyzed studies per country: Turkey and South Korea were the most represented countries. Interestingly, it is possible to identify the characteristic spread of BD studies along the ancient Silk Road.

### 3.2. Biomarkers and Their Roles in Diagnosis and Disease Activity

A total of 110 studies (99%) investigated serological biomarkers, while only two (2%) tested their population with urinary biomarkers. Most of the included studies (103; 93%) were designed to investigate the diagnostic potential of the biomolecules, while 62 (56%) tested their ability to differentiate between different stages of disease activity. A comprehensive view of all the examined markers is given in Table 2. The most important are cited in the following paragraphs.

#### 3.2.1. Conventional Inflammation Markers and Soluble Proteins

The erythrocyte sedimentation rate (ESR) and C reactive protein (CRP), two inflammation indices, have been assessed by 14 (13%) and 19 (17%) studies on BD, respectively. An increase in their values has been reported with a total agreement rate among the articles.

The neutrophil-to-lymphocyte ratio (NLR) is a parameter analyzed through a hemocytometer. It has been investigated as a biomarker in seven (6%) articles; all the studies reported a significant increase in the NLR in BD patients, especially in patients with active disease. 

Tumor necrosis factor-alpha (TNF-α) is a cytokine that regulates the immune system, inflammatory response, and apoptosis. Serum TNF-α has been analyzed in eight studies (7%). Its levels were remarkably increased in BD patients compared to healthy controls, whereas there have been inconclusive results on the correlation between high TNF-α levels and BD activity [29,49,58,61,69,81,103,117].

In the sub-group of interleukins (ILs), fifteen different molecules have been studied as serological markers in 22 studies (20%). Among them, IL-8 had increased levels in BD sera compared to controls in three different studies (3%) [50,51,75], with high rates of sensibility and specificity in differentiating active and inactive patients, as reported in four articles (4%) [19,36,50,51].Moreover, IL-6 has been investigated as a serological marker in six articles (5%), highlighting a potential in BD diagnosing but not in the activity disease classification. 

Adenosine deaminase (ADA) has been the main focus of three studies (3%). ADA is a marker of T-lymphocyte activation, whose serological levels were found to be markedly elevated in BD patients compared to controls [28,34,40].

Anti-alpha enolase antibodies (AAEA) have been evaluated in three studies (3%). They consist of a heterogeneous group of antibodies directed toward surface proteins in endothelial cells, which have been found to increase in many inflammatory diseases, including SLE, AR, and vasculitis. Additionally, in this case, the serological levels of both IgG and IgM AAEA seemed to be significantly elevated in BD patients, particularly during the active phase [60,92,107].

#### 3.2.2. Oxidant and Anti-Oxidant Molecules

Reactive oxygen species (ROS), including nitric oxide (NO), are products of oxidative stress and are usually released in inflammatory sites by the innate immune response and endothelial cells. In seven articles (6%), the authors described the NO levels to be significantly enhanced in the serum and urine of BD patients compared to HC [24,26,27,29,32,102]. Significant differences were noticed in patients with active disease in comparison to inactive patients [24,26,27,102,122].

Malondialdehyde (MDA), one of the final products of lipid peroxidation triggered by the free radicals of oxidative stress, was reported to be elevated in BD sera in comparison to HC, even if it was not a promising biomarker of BD activity of disease [38,56,102].

Super oxide dismutase (SOD) and catalase are anti-inflammatory enzymes involved in oxidative stress. The dosages of SOD and RBC catalase levels [28,102] showed significant reductions in BD patients, especially in samples collected during the phase of disease activity.

#### 3.2.3. microRNAs

Several miRNAs (including miR-93, miR-106b, miR-25, miR-146a, miR-326, and miR-181b) were assessed by three studies (3%) included in this systematic review. In particular, in a case-control study of 47 BD patients [103], the authors observed that the miR-155 levels increased in BD patients compared to HC. However, these results were not corroborated by the authors of two other studies [104,106], where a conspicuous decrease in miR-155 levels was conversely observed when testing their BD patients.

#### 3.2.4. New “Omics” Sciences

Two studies (2%) have addressed the serum metabolomics state of BD patients, starting with an untargeted approach and subsequently validating a specific panel of biomarkers on an independent cohort.

Through the gas chromatography/time-of-flight mass spectrometry GC/TOF-MS, Ahn and colleagues isolated a panel of five metabolites (decanoic acid, fructose, tagatose, linoleic acid, and oleic acid) able to differentiate BD patients from HC with high sensitivity and specificity, at 100% and 97.1%, respectively [125]. Concurrently, Zheng et al. observed that high serum levels of two polyunsaturated fatty acids (PUFAs), linoleic acid (LA) and arachidonic acid (AA), discriminated BD patients and HC efficiently with high sensitivity (95% for PUFAs and 95% for LA) and specificity (65 for PUFAs and 88% for LA) [110].

In their previous work, Ahn and colleagues also assessed the urinary metabolomic profiles of BD patients. The authors identified a combination of metabolites (guanine, pyrrole-2-carboxylate, 3-hydroxypyridine, mannose, L-citrulline, galactonate, isothreonate, sedoheptuloses, hypoxanthine, and gluconic acid lactone) able to identify BD patients with high sensitivity (96.7%) and specificity (93.3%) [93].

Two studies (2%) have widely investigated the serum proteomic asset using matrix-assisted laser desorption ionization–time-of-flight mass spectrometry (MALDI-TOF-MS). A first model based on 39 proteins could distinguish BD and HC with a sensitivity of 83.67% and a specificity of 89.87% [65]. The second study detected significant upregulation of fibrin, apolipoprotein A-IV, and serum amyloid A (SAA) in the sera of BD patients with active disease at the intestinal level compared to controls [94].

## 4. Discussion

BD is a rare multisystemic vasculitis whose symptoms and signs often overlap with other autoimmune diseases, leading to delayed diagnosis and occasionally inappropriate therapy. The pathogenesis of BD has not been fully elucidated. However, the dysregulation of the innate and acquired immune systems in a facilitative environment plays a crucial role in disease development [2] (Figure 4).

Further, unlike other autoimmune diseases, such as systemic lupus erythematosus (SLE), rheumatoid arthritis (AR), or other vasculitis, specific biomarkers for BD have not yet been identified, negatively affecting the early diagnosis and management of BD patients.

In this systematic review, we carefully reviewed all the relevant articles published in the current literature to identify the international efforts made in identifying specific serological and urinary BD biomarkers.

Considering the well-known inflammatory nature of BD, most studies have shown an increase in inflammatory biomarkers in BD patients, such as CRP, ESR, and numerous cytokines, including TNFα, IL-1β, IL-6, IL-8, IL-17, and IL-23 (Table 2). 

Unfortunately, despite a high agreement rate among the articles, their lack of specificity makes them a nonoptimal diagnostic tool, whereas they can be helpful in disease monitoring.

To date, there is consensus on the involvement of lymphocytes in the BD pathogenesis, in particular, T helper cells that produce IL-17 (Th17) and T regulatory (Tregs) cells [106,126,127]. In the presence of IL-23, Th naive cells differentiate in the Th17 phenotype and migrate at mucosal surfaces, where, through the secretion of IL-17, they induce the recruitment of neutrophils and activate epithelial cells, mediating the inflammatory process [128]. Conversely, Treg cells play a role in inhibiting the immune response triggered by the resident microflora in the mucosa by the secretion of TGF-β and IL-10 [129]. Interestingly, Th17 and Tregs share common pathways during differentiation, and their cell count is fundamental for maintaining the balance between pro-inflammatory and anti-inflammatory conditions in mucosal tissues [128]. Multiple studies on BD have reported a decrease in Treg cells, alongside an upregulation of Th17 cells and neutrophils (both as an absolute value and NLR) [76,80,84,87,106,116,122,123]. Activated neutrophils may reach the inflammatory sites, triggering a substantial oxidative stress response by releasing ROS, which can contribute to the disease progression over time. It is worth mentioning that increased levels of numerous pro-oxidants were observed in BD sera and urines; among them, ADA NO, advanced oxidation protein products (AOPPs), and some of the final products of lipid peroxidation, such as MDA and thiobarbituric acid-reactive substances (TBARS) [28,31,34,40,102] (Table 2). On the contrary, many studies have described lower anti-oxidant levels in the sera of BD patients, such as catalase and SOD, confirming the dysregulation of the production of pro-oxidants and anti-oxidant substances in BD [28,102]. However, similarly to the inflammation biomarkers, pro- and anti-oxidants remain aspecific and could be used for monitoring BD patients but, due to their low specificity, do not have a pivotal role in BD diagnosis.

MiRNAs are small non-coding RNA (19-23-nucleotide length) that inhibit translation by binding mRNAs. Recently, some miRNAs have been investigated as putative BD markers. In particular, low serum levels of miR-155 were detected in active BD patients, and higher levels were observed during disease remission [104,106]. It is known that miR-155 is involved in switching off the inflammatory response by downregulating IL-6 and IL-1β and upregulating IL-10, an inhibitory interleukin [130]. In fact, high serum levels of IL-6 and IL-1β and low levels IL-10 were observed in active BD [14,75,102,106,115,122]. 

Moreover, the role of miR-155 in blocking BD progression was confirmed by the increase in Th17 and the release of IL-17. These mechanisms are mediated by the inhibition of E26 transformation-specific-1 (ETS-1), a gene upregulated in BD [106,115,131]. One could hypothesize that lower levels of miR-155 might lead to low CD4+ T cells, Th17, and IL17 and increased ETS-1 during active BD [131]. However, Kolhai et al. reported an increase in the miR-155 levels in BD patients, in concomitance with a reduction of Ets-1 and an elevation of Th17 cells, suggesting a pro-inflammatory role and a potential therapeutic target [103]. While the current scientific interest is focused on miRNAs for improving our understanding of BD pathogenesis, their potential in diagnostic testing for BD remains to be elucidated.

Considering that BD disease is often described as an ensemble of phenotypes with different clinical characteristics, a future challenge could be to test if these phenotypes exhibit different miRNA patterns [132]. This could not only improve our knowledge about pathogenic processes underlying the various phenotypes but could also represent a step toward a more tailored therapeutic approach.

To date, new “omics“ science, such as proteomics and metabolomics, has provided a comprehensive analysis of endogenous proteins and metabolites. With the use of metabolomics, one can potentially detect the alterations of physiological and pathological metabolites at the early stages of the disease due to its excellent sensitivity. In BD, two metabolomic tests have been developed and subsequently validated with reported high specificity and sensitivity [110,125]. Unfortunately, although this approach seems to be very promising, these tests are extremely expensive and complex, and therefore, are far from being routinely available for diagnostic or follow-up testing.

In addition to metabolome investigations from blood samples, many studies have recently focused on analyzing the fecal metabolome alterations, resulting from changes in the gut microbial communities in BD patients [133,134]

Since the intercorrelation between diet and gut microbiota is well known, studying intestinal altered metabolic profiles and the microbial community imbalance of BD patients is paving the way to new therapeutic approaches based on nutritional interventions [135]. 

We acknowledge that this study suffers from some limitations, mainly due to the vast heterogeneity of the included studies regarding the number of patients, control groups, and the types of biomarkers and assays used. Moreover, although it was possible to include only BD diagnosed following the ISGBD criteria, there was no standard disease activity score in the past. Only recently has a consensus on a common definition of Behçet’s disease activity been reached by developing and validating the Behçet’s Disease Current Activity Form (BDCAF) score [124]; however, it is not objectionable because it is a subjective score based on referred symptoms. For these reasons, a meta-analysis of the studies could not be performed.

## 5. Conclusions

In conclusion, despite the enormous efforts from the scientific community to identify potential biomarkers in BD, much more work must be done. While identifying novel aspecific biomarkers might help us better understand BD’s pathogenesis and might also find a place for monitoring disease activity during follow-up, we are still far from identifying potential diagnostic biomarkers for this complex and rare disease. Proteomics, metabolomics, and microbiome analysis might help in the near future to identify potential candidates to help researchers and ultimately clinicians to better identify patients suffering from BD.

## Figures and Tables

**Figure 1 ijms-24-03041-f001:**
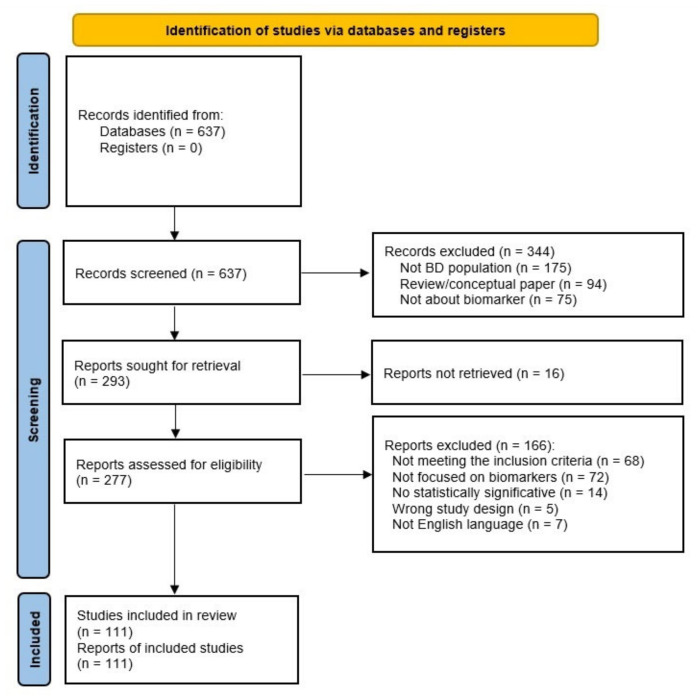
Flowchart of the literature search strategy.

**Figure 2 ijms-24-03041-f002:**
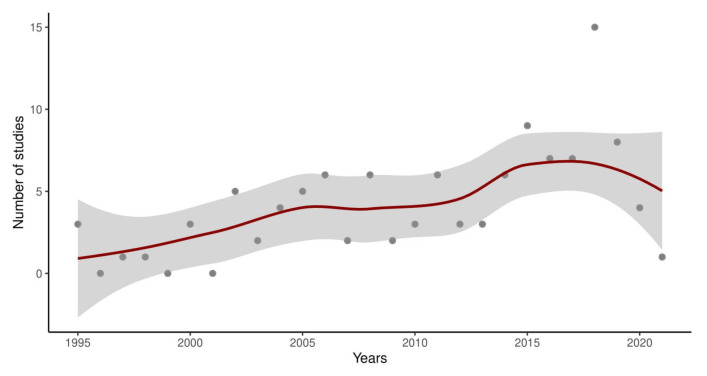
Graphical representation of the number of studies per year included in this systematic review. The scatter plot was established using the package Ggplot2 [12] of R studio [13].

**Figure 3 ijms-24-03041-f003:**
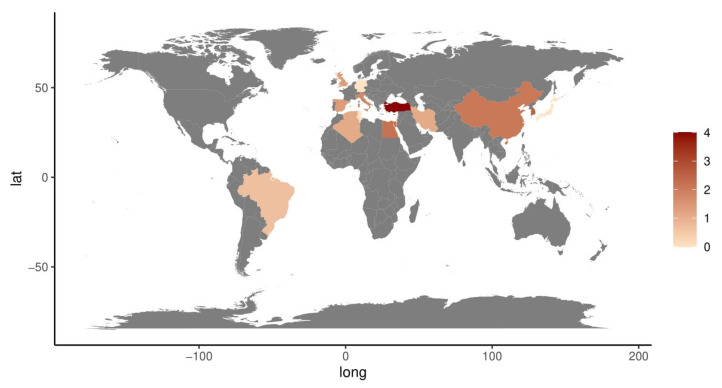
Graphical representation of the global origin of the publication rate of the analyzed studies per country, colored by BD study rate. The graphical representation was computed by log-transforming the number of research papers published by each country. It is possible to recognize the Silk Road pattern. The map was created using the package Ggplot2 [124] of R studio [12].

**Figure 4 ijms-24-03041-f004:**
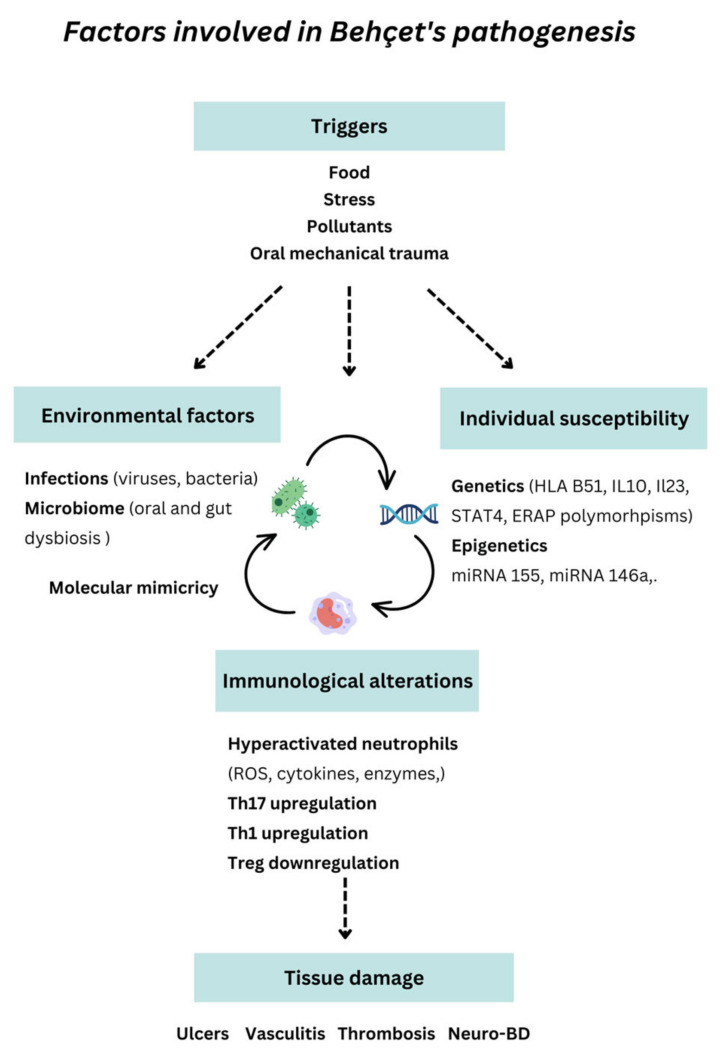
Mechanisms underlying Behçet’s disease’s etiopathogenesis.

**Table 1 ijms-24-03041-t001:** Main characteristics of the studies included in the analysis.

REF	Year	First Author	Country	Design	BD Patients, n	Controls, n	Biomarker Tested	Urinary/ Serologic	Diagnostic/Activity
[14]	1995	Yosipovitch et al.	Israel	Retrospective	25	20	IL-1B	S	Diagnostic
							SIL-2R	S	Diagnostic
[15]	1995	Deǧer et al.	Turkey	Retrospective	42 (20 active)	40	PMN elastase	S	Diagnostic/Activity
[16]	1995	Direskeneli et al.	UK	Retrospective	70 (56 active)	52	AECA	S	Diagnostic/Activity
							vVF	S	Diagnostic
[17]	1997	Uslu et al.	Turkey	Retrospective	27	18	ET-1	S	Diagnostic
[18]	1998	Alpsoy et al.	Turkey	Retrospective	32 (14 active)	20	IL-2	S	Diagnostic
							SIL-2R	S	Activity
[19]	2000	Katsantonis et al.	Germany	Retrospective	34 (25 active)	N/A	IL-8	S	Activity
[20]	2000	Eksioglu-Demiralp et al.	Turkey	Retrospective	37	55	CD4+CD16+	S	Diagnostic
							CD4+CD56+	S	Diagnostic
[21]	2000	Freysdottir et al.	UK	Retrospective	20	26	T-γδ	S	Diagnostic
							CD56	S	Diagnostic
[22]	2002	Krause et al.	Israel	Retrospective	27	20	IgG ASCA	S	Diagnostic
							IgA ASCA	S	Diagnostic
[23]	2002	Evereklioglu et al.	Turkey	Retrospective	35 (18 active)	20	Leptin	S	Diagnostic/Activity
[24]	2002	Er et al.	Turkey	Retrospective	43 (20 active)	52	ET-1	S	Diagnostic/Activity
							Homocysteine	S	Diagnostic/Activity
							NO	S	Diagnostic/Activity
[25]	2002	Saglam et al.	Turkey	Retrospective	44 (23 active)	30	cICAM-1	S	Diagnostic/Activity
[26]	2002	Evereklioglu et al.	Turkey	Retrospective	52 (27 active)	32	NO	S	Diagnostic/Activity
[27]	2003	Evereklioglu et al.	Turkey	Retrospective	36 (16 active)	20	NO (urinary)	U	Diagnostic/Activity
							NO (serum)	S	Diagnostic/Activity
[28]	2003	Erkiliç et al.	Turkey	Retrospective	35 (17 active)	20	ADA	S	Diagnostic/Activity
							TBARS	S	Diagnostic
							Plasmatic SOD	S	Diagnostic/Activity
							RBC SOD	S	Diagnostic/Activity
							Plasmatic GSHPx	S	Diagnostic/Activity
							RBC GSHPx	S	Diagnostic/Activity
							RBC Catalase	S	Diagnostic
[29]	2004	Akdeniz et al.	Turkey	Retrospective	27	16	IL-6	S	Diagnostic
							Il-2	S	Diagnostic
							TNF-α	S	Diagnostic
							NO	S	Diagnostic
[30]	2004	Sari et al.	Turkey	Retrospective	23	20	E-selectine	S	Diagnostic
							ESR	S	Diagnostic
							PCR	S	Diagnostic
[31]	2004	Yazici et al.	Turkey	Retrospective	49 (31 active)	40	MPO	S	Diagnostic/Activity
							AOPP	S	Diagnostic/Activity
							Thiol	S	Diagnostic/Activity
[32]	2004	Duygulu et al.	Turkey	Retrospective	23 (11 active)	15	NO	S	Diagnostic/Activity
[33]	2005	Ureten et al.	Turkey	Retrospective	72 (37 active)	73	CD64	S	Diagnostic/Activity
[34]	2005	Calis et al.	Turkey	Retrospective	75 (50 active)	25	ADA	S	Diagnostic/Activity
[35]	2005	Qiao et al.	Japan	Retrospective	35 (15 active)	16	CXCR2	S	Diagnostic/Activity
[36]	2005	Gür-Toy et al.	Turkey	Retrospective	67	0	IL-8	S	Activity
							CRP	S	Diagnostic
							ESR	S	Diagnostic
[37]	2005	Coskun et al.	Turkey	Retrospective	40 (25 active)	30	Neopterin	S	Diagnostic/Activity
							ESR	P	Diagnostic/Activity
							CRP	S	Diagnostic/Activity
[38]	2006	Yardim-Akaydin et al.	Turkey	Retrospective	23	43	Allantoin	S	Diagnostic
							MDA	S	Diagnostic
							Ascorbic acid	S	Diagnostic
[39]	2006	Kose et al.	Turkey	Retrospective	68 (51 active)	17	Neopterin	S	Diagnostic/Activity
[40]	2006	Canpolat et al.	Turkey	Retrospective	23 (10 active)	20	ADA	S	Diagnostic/Activity
							Erythrocyte ADA	S	Diagnostic/Activity
[41]	2006	Kwon et al.	South Korea	Prospective	211 (92 active)	N/A	Protein S	S	Activity
[42]	2006	Briani et al.	Italy	Retrospective	32	118	Anti-HS igM	S	Diagnostic
							Anti-HS igG	S	Diagnostic
[43]	2006	Sarican et al.	Turkey	Retrospective	64 (25 active)	26	Homocysteine	S	Diagnostic/Activity
[44]	2007	Lee et al.	South Korea	Retrospective	50 (26 active)	UK	Gal-3	S	Diagnostic/Activity
							G3BP	S	Activity
[45]	2007	Pay S et al.	Turkey	Retrospective	58 (23 active)	20	MMP-2	S	Diagnostic
							MMP-9	S	Diagnostic/Activity
[46]	2008	Öztürk et al.	Turkey	Retrospective	21	21	VEGF	S	Diagnostic
							ESR	S	Diagnostic
							CRP	S	Diagnostic
[47]	2008	Turan et al.	Turkey	Prospective	35	N/A	sTNFR1	S	Activity
							sTNFR2	S	Activity
[48]	2008	Kutlay et al.	Turkey	Retrospective	45 (33 active)	15	CEC	S	Diagnostic/Activity
[49]	2008	Curnow et al.	UK	Retrospective	52 (24 active)	35	IL-15	S	Diagnostic/Activity
							CXCL-8	S	Diagnostic/Activity
							TNF-α	S	Diagnostic/Activity
[50]	2008	Polat et al.	Turkey	Retrospective	32	16	IL-8	S	Diagnostic/Activity
[51]	2008	Durmazlar et al.	Turkey	Retrospective	45 (33 active)	29	IL-8	S	Diagnostic/Activity
[52]	2009	Habibagah et al.	Iran	Retrospective	53 (15 active)	44	IL-23	S	Diagnostic/Activity
							E–cadherin	S	Diagnostic
[53]	2009	Fadini et al.	Italy	Retrospective	30	27	CD34+KDR+ EPCs	S	Diagnostic
							CD34+CD133+KDR+ EPCs	S	Diagnostic
[54]	2010	Choe et al.	South Korea	Retrospective	59 (21 active)	65	Angiopoietin-1	S	Diagnostic
							Angiopoietin-2	S	Diagnostic
[55]	2010	Donmez et al.	Turkey	Retrospective	89 (17 active)	86	aTAFI	S	Diagnostic
							Thrombomodulin		Diagnostic
[56]	2010	Sezer et al.	Turkey	Retrospective	60 (33 active)	46	MDA	S	Diagnostic
							8-OHdG	S	Diagnostic/Activity
							T-SH	S	Diagnostic
[57]	2011	Özden et al.	Turkey	Retrospective	70	61	Gal-3	S	Diagnostic/Activity
[58]	2011	Pehlivan et al.	Turkey	Retrospective	45 (25 active)	30	Resistin	S	Diagnostic/Activity
							TNF-α	S	Diagnostic/Activity
[59]	2011	Ahn et al.	South Korea	Retrospective	71 (21 active)	34	α defensin1	S	Activity
							αdefensin1 mRNA	S	Diagnostic/Activity
[60]	2011	Shin et al.	South Korea	Retrospective	80	23	AAEA	S	Diagnostic
[61]	2011	Jung et al.	South Korea	Retrospective	88 (30 severe, 12 moderate)	10	sTREM1	S	Diagnostic/Activity
							TNF-α		Diagnostic
[62]	2011	Vural et al.	Turkey	Retrospective	20	40	STIP-1	S	Diagnostic
[63]	2012	Bello et al.	Spain	Retrospective	30	28	sCD40L	S	Diagnostic
							MMP-9	S	Diagnostic
[64]	2012	Gündüz et al.	Turkey	Retrospective	40 (11 active)	20	CD4+CD25+FOXP3+Treg	S	Diagnostic/Activity
							CD4+FOXP3+Treg	S	Diagnostic/Activity
[65]	2012	Wang et al.	China	Retrospective	49	79	Proteomic analysis	S	Diagnostic
[66]	2013	Örem et al.	Turkey	Retrospective	72 (40 active)	30	Lipoprotein-associated phospholipase A2	S	Diagnostic/Activity
							CRP	S	Diagnostic/Activity
							ESR	S	Diagnostic/Activity
[67]	2013	Hamzaoui et al.	Tunisia	Retrospective	46 (20 active)	70	IL-33	S	Diagnostic/Activity
							IL6	S	Diagnostic
							IL7	S	Diagnostic
[68]	2013	Vural et al.	Turkey	Retrospective	144	168	MTCH1 Ab	S	Diagnostic
[69]	2014	Shaker et al.	Egypt	Retrospective	30 (20 active)	20	TNF- α	S	Diagnostic/Activity
							APRIL	S	Diagnostic/Activity
							BCMA	S	Diagnostic/Activity
							BAFF	S	Diagnostic/Activity
							CRP	S	Diagnostic/Activity
							ESR	S	Diagnostic/Activity
[70]	2014	Xun et al.	China	Retrospective	58	106	Prohibitin	S	Diagnostic
[71]	2014	Vayà et al.	Spain	Retrospective	89	94	RDW	S	Diagnostic
							CRP	S	Diagnostic
							Fibrinogen	S	Diagnostic
							Leucocytes	S	Diagnostic
							Neutrophils	S	Diagnostic
[72]	2014	Balta et al.	Turkey	Retrospective	33 (16 active)	35	Endocan	S	Diagnostic/Activity
							CRP	S	Diagnostic
							ESR	S	Diagnostic
[73]	2014	Ozuguz et al.	Turkey	Prospective	40	20	ADMA	S	Diagnostic
							CRP	S	Diagnostic/Activity
							ESR	S	Diagnostic/Activity
							Homocysteine	S	Diagnostic/Activity
[74]	2014	Mejia et al.	Spain	Prospective	56 (17 active)	56	Prothrombin fragm. 1.2	S	Diagnostic/Activity
							Factor VIII	S	Diagnostic/Activity
							vWF	S	Diagnostic
[75]	2015	Lopalco et al.	Italy	Prospective	58	32	IL-6	S	Diagnostic
							IL-8	S	Diagnostic
							IL-18	S	Diagnostic
							IFN-α	S	Diagnostic
							CXCL11	S	Diagnostic
[76]	2015	Yuksel et al.	Turkey	Retrospective	36 (17 active)	35	ADMA	S	Diagnostic/Activity
							NLR	S	Diagnostic/Activity
[77]	2015	Bassyouni et al.	Egypt	Retrospective	47	30	Angiopoietin-1	S	Diagnostic
[78]	2015	Tulunay et al.	Turkey	Retrospective	26	26	STAT3	S	Diagnostic
[79]	2015	Belguendouz et al.	Algeria	Retrospective	26 (16 active)	17	IL-18	S	Activity
[80]	2015	Ozturk et al.	Turkey	Retrospective	65 (40 active)	62	NLR	S	Diagnostic/Activity
[81]	2015	Turkcu et al.	Turkey	Retrospective	51 (25 active)	24	TNF-α	S	Diagnostic
							Resistin	S	Diagnostic
							Omentin	S	Diagnostic
[82]	2015	De Souza et al.	Brazil	Retrospective	26 (13 active)	20	HMGB1	S	Diagnostic
[83]	2015	Seo et al.	South Korea	Retrospective	112 (66 active)	45	YKL-40	S	Diagnostic/Activity
[84]	2016	Yolbas et al.	Turkey	Retrospective	53 (6 active)	55	NLR	S	Activity
						91			
						51+39			
[85]	2016	Hu et al.	China	Retrospective Phase I	40 (identification)	35	Protein microarray		
				Phase II	130 (validation)	223	Anti-CTDP1 Ab	S	Diagnostic
[86]	2016	Mejia et al.	Spain	Retrospective	55	73	Procoagulant microparticles	S	Diagnostic
[87]	2016	Balkarli et al.	Turkey	Retrospective	186 (120 active)	79	NLR	S	Diagnostic
							ESR	S	Diagnostic/Activity
							CRP	S	Diagnostic
[88]	2016	Park et al.	South Korea	Retrospective	51 (29 active)	N/A	Anti-lysozyme	S	Activity
[89]	2016	Cantarini et al.	Italy	Retrospective	27 (57 total samples: 21 from active, 36 inactive)	36	CD40L	S	Diagnostic
							Leptin	S	Diagnostic
							sTNFR	S	Diagnostic
							IL-6	S	Diagnostic
							ESR	S	Activity
[90]	2017	Cure et al.	Turkey	Retrospective	84	84	AIP	S	Diagnostic/Activity
							CRP	S	Diagnostic
[91]	2017	Jiang et al.	China	Retrospective	140 (108 active)	107	PLR	S	Diagnostic/Activity
							LMR	S	Diagnostic
							ESR	S	Activity
							CRP	S	Activity
[92]	2017	Kang et al.	South Korea	Retrospective	110	110	AAEA IgG	S	Diagnostic
[93]	2017	Ahn JK et al.	South Korea	Retrospective	44	41	Panel of 10 urinary biomarkers: guanine, pyrrole-2-carboxylate, 3-hydroxypyroline, mannose, L-citrulline, galactonate, isothreonate, sedoheptulose, hypoxanthine, and gluconic acidlactonate	U	Diagnostic
							Guanine	U	Diagnostic
							Pyrrole-2-carboxylate	U	Diagnostic
							3-hydroxypyroline	U	Diagnostic
							Mannose	U	Diagnostic
							L-citrulline	U	Diagnostic
							Galactonate	U	Diagnostic
							Isothreonate	U	Diagnostic
							Sedoheptulose	U	Diagnostic
							Hypoxanthine	U	Diagnostic
							Gluconic acidlactonate	U	Diagnostic
[94]	2017	Lee et al.	South Korea	Retrospective Phase I	15 (identification)	15	Fibrin, apoliprorotein A-IV and SAA	S	Diagnostic
				Phase II	49 (validation)	41	SAA	S	Diagnostic
							IL-1β	S	Diagnostic
[95]	2017	Ha et al.	South Korea	Retrospective	50 (29 active)	35	IL-32	S	Diagnostic
[96]	2017	Lopalco et al.	Italy	Retrospective	46	19	sTNFR1	S	Diagnostic
							sTNFR2	S	Diagnostic
							Chitinase3-like1	S	Diagnostic
							gp130/sIL-6Rb	S	Diagnostic
							IL-26	S	Diagnostic
[97]	2018	Omma et al.	Turkey	Retrospective	93 (57 active)	62	Calprotectin	S	Diagnostic
							CRP	S	Diagnostic
							IMA	S	Diagnostic
[98]	2018	Koca et al.	Turkey	Retrospective	71	75	Bilirubin	S	Diagnostic
[99]	2018	Enecik et al.	Turkey	Retrospective	45 (28 active)	25	IL-20	S	Diagnostic
[100]	2018	Harmanci et al.	Turkey	Retrospective	30	30	VEGF gene expression levels	S	Diagnostic
[101]	2018	Lucherini et al.	Italy	Retrospective	72	29	IgD	S	Diagnostic
[102]	2018	Chekaoui et al.	Algeria	Retrospective	48 (28 active)	41	IL-1β	S	Diagnostic/Activity
							NO	S	Diagnostic/Activity
							AOPP	S	Diagnostic/Activity
							MDA	S	Diagnostic
							SOD	S	Diagnostic/Activity
[103]	2018	Kolahi et al.	Iran	Retrospective	47	61	mir-155	S	Diagnostic
							TNF-α expression	S	Diagnostic
[104]	2018	Ahn et al.	South Korea	Retrospective	45	45	Panel of 5 biomarkers: DA, fructose, tagatose, LA, and OA	S	Diagnostic
[105]	2018	Saylam et al.	Turkey	Retrospective	30	41	suPAR	S	Diagnostic
							CRP	S	Diagnostic
[106]	2018	Ahmadi et al.	Iran	Retrospective	47	58	Th17	S	Diagnostic
							Treg	S	Diagnostic
							RORɣt mRNA	S	Diagnostic
							FoxP3 mRNA	S	Diagnostic
							IL-17mRNA	S	Diagnostic
							IL-23 mRNA	S	Diagnostic
							TGF mRNA	S	Diagnostic
							IL-10 mRNA	S	Diagnostic
							IL-17	S	Diagnostic
							IL-23	S	Diagnostic
							IL-10	S	Diagnostic
							TFG-beta	S	Diagnostic
							miR-93	S	Diagnostic
							miR-106b	S	Diagnostic
							miR-25	S	Diagnostic
							miR-146°	S	Diagnostic
							miR-155	S	Diagnostic
							miR-326	S	Diagnostic
[104]	2018	Hassouna et al.	Egypt	Retrospective	30	15	miR-155	S	Diagnostic
[107]	2018	Prado et al.	Brazil	Retrospective	97 (43 active)	123	AAEA IgM	S	Diagnostic/Activity
[108]	2018	Acikgoz et al.	Turkey	Retrospective	60	50	MHR	S	Diagnostic
[109]	2018	Hasan et al.	UK	Retrospective	60 (44 active)	60	NK	S	Diagnostic
							CD56Dim	S	Diagnostic
							CD56Brigh	S	Diagnostic
[110]	2018	Zheng et al.	China	Retrospective Phase I	24 (identification)	26	PC (34:3)	S	Diagnostic
							PC (40:8)	S	Diagnostic
							LA	S	Diagnostic
							AA	S	Diagnostic
				Phase II	25 (validation)	19	LA	S	Diagnostic
						27	AA	S	Diagnostic
[111]	2019	Şahin et al.	Turkey	Retrospective	46	44	Pannexin-1	S	Diagnostic
[112]	2019	Bassyouni et al.	Egypt	Retrospective	87	60	CCN2	S	Diagnostic
[113]	2019	Arica et al.	Turkey	Retrospective	45 (32 active)	28	Early EPCs	S	Diagnostic/Activity
							Late EPCs	S	Diagnostic/Activity
							MMP9	S	Diagnostic
							VEGF	S	Diagnostic/Activity
							CRP	S	Diagnostic/Activity
							ESR	S	Diagnostic
[114]	2019	Sandikci et al.	Turkey	Retrospective	150	100	Serumnativethiol	S	Diagnostic
							Total thiol	S	Diagnostic
							T-SH	S	Diagnostic
[115]	2019	Talaat et al.	Egypt	Retrospective	64	20	IL-6	S	Diagnostic/Activity
							IL-10	S	Diagnostic
							IL-17	S	Diagnostic
[116]	2019	Gheita et al.	Egypt	Retrospective	96	60	NLR	S	Diagnostic
							PLR	S	Diagnostic
							RDW	S	Diagnostic
							MPV	S	Diagnostic
							VEGF	S	Diagnostic
[117]	2019	El Boghdady et al.	Egypt	Retrospective	51	45	TNF-α	S	Diagnostic
							IL-6	S	Diagnostic
							E-selectine	S	Diagnostic
							VCAM	S	Diagnostic
							miR-181b	S	Diagnostic
[118]	2019	Balbaba et al.	Turkey	Retrospective	48 (24 active)	24	Cortistatin	S	Diagnostic
[119]	2020	Hassan et al.	Egypt	Retrospective	42	42	Endocan	S	Diagnostic/Activity
[120]	2020	Hussain et al.	China	Retrospective	50	100	Moesin	S	Diagnostic
[121]	2020	Hussain et al.	China	Retrospective	32	64	NuMA Ab	S	Diagnostic
[122]	2020	Djaballah-Ider et al.	Algeria	Retrospective	61 (47 active)	25	NLR	S	Activity
							NO	S	Activity
							IL-4	S	Activity
							IFN-gamma	S	Activity
[123]	2021	Cheng et al.	China	Retrospective	48 (34 active)	96	Lymphocyte count	S	Diagnostic/Activity
							White blood cell count	S	Diagnostic
							Neutrophil count	S	Diagnostic
							Basophil count	S	Diagnostic/Activity
							RDW	S	Diagnostic/Activity
							MCH	S	Diagnostic
							MCHC	S	Diagnostic
							Platelet count	S	Diagnostic/Activity
							Plateletcount	S	Diagnostic/Activity
							MPV	S	Diagnostic/Activity
							CRS	S	Diagnostic
							PLR	S	Diagnostic/Activity
							NLR	S	Diagnostic
							Monocyte	S	Activity
							LMRcount	S	Diagnostic

8-OHdG—8-hydroxy-2′-deoxyguanosine; AA—arachidonic acid; AAEA—anti-alpha-enolase antibodies; ADA—adenosine deaminase; ADMA—asymmetric dimethyl arginine; AECA—anti-endothelial cell antibodies; AIP—atherogenic index plasma, anti-HS—anti-heparin–sulfate antibodies; anti-CTDP1—anti-carboxy-terminal domain phosphatase subunit 1; AOPP—advanced oxidation protein products; APRIL—a proliferation-inducing ligand; ASCA—anti-Saccharomyces cerevisiae; aTAFI—activated thrombin activatable fibrinolysis inhibitor; BAFF—B-cell-activating factor; BCMA—B-cell maturation antigen; CEC- circulating endothelial cells; cICAM—circulating intercellular adhesion molecule-1; cNuM—anuclear mitotic apparatus protein located at the carboxyl terminus; CPR—C-reactive protein; CTGF—connective tissue growth factor;CXCL11—C-X-C motif chemokine 11; CXCR2—C-X-C motif chemokine receptor 2; DA—decanoic acid; Endocan—human endothelial cell-specific molecule-1; EPC—endothelial progenitor cells; ESR—erythrocyte sedimentation rate; ET-1—endothelin-1; ETP—endogenous thrombin potential; GAL-3—galectin-3; G3BP—galectin-3 binding protein; HMGB1—high-mobility group box 1; IgD—D immunoglobulin;IMA—ischemia-modified albumin; INFa—interferon alpha; INFg—interferon gamma; LA—linoleic acid; LMR—lymphocytes-to-monocytes ratio; LpPLA2—lipoprotein-associated phospholipase A2; MDA—manoldialdehyde; MHR—monocyte-to-high-density lipoprotein–cholesterol ratio; MMP—matrix metalloproteinase; MPO—plasma myeloperoxidase; MPV—mean platelet volume; MTCH1—mitochondrial carrier homolog 1; NLR—neutrophil-to-lymphocyte ratio; NO—nitric oxide; OA—oleic acid; PC—phosphatidylcholines; PLR—platelet-to-lymphocyte ratio; PMN—polymorph nuclear; Procoagulant MP—procoagulant microparticles; RDW—red cell distribution width; SAA—serum amyloid A; SIL-1R—Soluble interleukin-1 receptor; SIL6-RB-Soluble interleukin-6 receptor B; SOD—Superoxide dismutase; STIP1—Stress induced phosphoprotein 1; sTNFR—soluble tumor necrosis factor receptor; sTREM1—soluble triggering receptor expressed on myeloid cells; suPAR—soluble urokinase plasminogen activator receptor; TBARS—thiobarbituric acid-reactive substances TGF-b—transforming growth factor beta; TNFa—tumor necrosis factor alpha; T-SH—total sulfhydryl levels; VCAM—vascular cell adhesion molecule 1;VEGF—vascular endothelial growth factor; vWF—von Willebrand factor.

**Table 2 ijms-24-03041-t002:** Serological and urinary biomarkers investigated in the studies included in the systematic review.

ILs	IL-1β [14,94,102]	IL-2 [18,29]	IL-4 [122]	IL-6 [29,67,75,89,115,117]	IL-7 [67]	IL-8 [19,36,49,50,51,75]	IL-10 [106,115]	IL-15 [49]	IL-17 [106,115]	IL-18 [75,79]	IL-20 [99]	IL-23 [52,106]
	IL-26 [96]	IL-32 [95]	IL-33 [67]									
Cytokines	TNF-α [29,49,58,61,69,81,103,117]	TGF-β [106]	APRIL [69]	BAFF [69]	INF-α [75]	IFN-γ [122]	CTGF [112]	STAT3 [78]	CXCL11 [75]			
Surface proteins	CD64 [33]	CXCR2 [35]	BMCA [69]	VCAM [117]								
Soluble proteins	SIL-2R [14,18]	PMN leukocyte elastase [15]	AECA [16]	vWF [16,74]	ET-1 [17,24]	Anti-ASCA Ab [22]	Leptin [23,89]	Homocysteine [24,43,73]	CRP [30,31,36,37,64,66,69,71,72,73,87,90,91,93,99,105,113,123]	cICAM-1 [23]	Catalase [28]	ADA [28,34,40]
	SOD [28,102]	TBARS [28]	E-selectine [30,117]	MPO [31]	Neopterin [37,39]	VEGF [46,100,113,116]	Protein S [41]	antiHS [42]	Gal-3 [44,57]	G3BP [44]	MMP2 [45]	α-defensin1 [59]
	sTNFR1 e 2 [47,89,96]	E-Caderin [52]	Angiopoietin1 [54,77]	Resistin [58,81]	Thrombomodulin [55]	aTAFI [55]	AAEA [60,92,107]	sTREM1 [61,100]	STIP [62]	sCD40L [63,89]	MMP9 [63,113]	Lp-PLA2 [66]
	MTCH1 Ab [68]	Prohibitin [70]	Endocan [72,119]	ADMA [73,76]	Omentin [81]	HMBG1 [82]	Anti-lysozyme [88]	Fibrinogen [71]	Factor VIII [74]	cNuMA Ab [121]	anti-CTDP1 Ab [85]	SAA [94]
	sIL6-RB [96]	Chitinase3-like1 [83,96]	Bilirubin [98]	Calprotectin [97]	IMA [97]	IgD [101]	suPAR [105]	Pannexin-1 [111]	Cortistatin [118]	Moesin [120]		
Cells	CD4+CD16+ [19,20]	CD4+CD56+ [19,20]	T γδ [21]	CEC [48]	CD34+KDR+EPCs [53,113]	CD34+CD133+KDR+ EPCs [53]	CD4+CD25+FOXP3+Treg [64]	CD4+FOXP3+Treg [64]	Treg [106]	Th17 [106]	CD56 + [109]	
miRNA	α-defensin 1 [59]	miR-155 [103,104,106]	miR-181b [117]	miR-93 [106]	miR-106b [106]	miR-25 [106]	miR-146a [106]	miR-326 [106]				
Metabolomic/ proteomic markers	DA, OA Fructose, tagatose [125]	LA [110,125]	PC [110]	AA [110]	Panel of six proteomic biomarkers [65]							
Others	ESR [30,31,36,37,46,52,66,69,72,87,89,91,99,113]	NO [24,26,27,29,32,102,122]	Thiol [31,114]	AOPP [31,102]	Allantoin [38]	MDA [38,56,102]	Ascorbic acid [38]	8-OhdG [56]	T-SH [56,114]	PLR [31,91,116]	LMR [91,123]	NLR [31,76,80,84,87,116,122]
	AIP [90]	RDW [71]	ETP [74]	MPV [123]	RDW [71,123]	Procoagulant MP [86]	MHR [108]					
Urinary markers	Metabolomic panel: Guanine Pyrrole-2-carboxylate 3-hydroxypyroline Mannose L-citrulline Galactonate Isothreonate Sedoheptulose Hypoxanthine Gluconic acidlactonate [93]		NO [14]									

8-OHdG—8-hydroxy-2′-deoxyguanosine; AA—arachidonic acid; AAEA—anti-alpha-enolase antibodies; ADA—adenosine deaminase; ADMA—asymmetric dimethyl arginine; AECA—anti-endothelial cell antibodies; AIP—atherogenic index plasma, anti-HS—anti-heparin–sulfate antibodies; anti-CTDP1—anti-carboxy-terminal domain phosphatase subunit 1; AOPP—advanced oxidation protein products; APRIL—a proliferation-inducing ligand; ASCA—anti-Saccharomyces cerevisiae; aTAFI—activated thrombin activatable fibrinolysis inhibitor; BAFF—B-cell-activating factor; BCMA—B-cell maturation antigen; CEC- circulating endothelial cells; cICAM—circulating intercellular adhesion molecule-1; cNuM—a nuclear mitotic apparatus protein located at the carboxyl terminus; CPR—C-reactive protein; CTGF—connective tissue growth factor; CXCL11—C-X-C motif chemokine 11; CXCR2—C-X-C motif chemokine receptor 2; DA—decanoic acid; Endocan—human endothelial cell-specific molecule-1; EPC—endothelial progenitor cells; ESR—erythrocyte sedimentation rate; ET-1—endothelin-1; ETP—endogenous thrombin potential; GAL-3—galectin-3; G3BP—galectin-3 binding protein; HMGB1—high-mobility group box 1; IgD—D immunoglobulin; IMA—ischemia-modified albumin; INFa—interferon alpha; INFg—interferon gamma; LA—linoleic acid; LMR—lymphocytes-to-monocytes ratio; LpPLA2—lipoprotein-associated phospholipase A2; MDA—manoldialdehyde; MHR—monocyte-to-high-density lipoprotein–cholesterol ratio; MMP—matrix metalloproteinase; MPO—plasma myeloperoxidase; MPV—mean platelet volume; MTCH1—mitochondrial carrier homolog 1; NLR—neutrophil-to-lymphocyte ratio; NO—nitric oxide; OA—oleic acid; PC—phosphatidylcholines; PLR—platelet-to-lymphocyte ratio; PMN—polymorph nuclear; Procoagulant MP—procoagulant microparticles; RDW—red cell distribution width; SAA—serum amyloid A; SIL-1R—Soluble interleukin-1 receptor; SIL6-RB-Soluble interleukin-6 receptor B; SOD—Superoxide dismutase; STIP1—Stress induced phosphoprotein 1; sTNFR—soluble tumor necrosis factor receptor; sTREM1—soluble triggering receptor expressed on myeloid cells; suPAR—soluble urokinase plasminogen activator receptor; TBARS—thiobarbituric acid-reactive substances TGF-b—transforming growth factor beta; TNFa—tumor necrosis factor alpha; T-SH—total sulfhydryl levels; VCAM—vascular cell adhesion molecule 1; VEGF—vascular endothelial growth factor; vWF—von Willebrand factor.

## Data Availability

Not applicable.

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
