# Peer review of "Finding the Needle in the Haystack: Serological and Urinary Biomarkers in Behçet’s Disease: A Systematic Review"

_ijms, 2023, doi:10.3390/ijms24033041_

Round 1
Reviewer 1 Report
Arbrile et al., describe Behcet’s disease. The literature search and presentation is well done. The reviewer realizes that that the etiology, cause and reliable symptoms and thus diagnosis is not clear. This is known but for an (interested) reader not really helpful.
However, there is a regional concentration of cases and at least some genetic associations (HLA) are known. For a specialist it might be possible to come up with a hypothesis as to why some of the facts would fit into a certain picture. In addition, the reader would like to know when the disease usually starts, concentrating perhaps on the maturity of the immune system around 20 – 25. Is there a female/male disproportion. This hypothesis – it is only a hypothesis – would provide a basis to perhaps test it. This could be put in a graphical picture.
If this is not possible, the review is merely a collection of papers nicely presented.
Author Response
Thank you for all the comments and suggestions. The aim of the study was to systemically review the available literature in order to identify possible serum or urinary biomarkers in Behçet’s disease.
We agree with the reviewer that the main take home message of our study is to highlight the vast heterogeneity existing in the field. With all its limitation, our manuscript has the strength to provide a comprehensive overview of the biomarker research in the field. We hope it might set the state of the art for future research. Having said that, the manuscript has been implemented with more information regarding F/M prevalence and age onset, to read: “Females and males are equally interested, with a worse disease progression in males due to ocular, vascular and neurological involvement [2]. Diagnosis onset is collocated between 25 and 30 years old, although countries with a low disease prevalence may show delayed time of diagnosis [3].”
In addition, a new figure has been included in the manuscript in order to give a reader an idea of the etiological mechanisms involved in BD pathogenesis.

Reviewer 2 Report
In this systematic review article, Arbrile et al. have tried to identify a biomarker for Behçet’s disease (BD). They analyzed available literature on serological and urinary BD biomarkers from the last 25 years. Major findings include an increase in inflammatory markers and in pro-oxidant molecules, controversial involvement of microRNA-155, and a new biomarker strategy by analyzing serum metabolomics via “omics sciences”. The sample size is a major concern for these “omics sciences “studies. Despite analyzing many studies, a proper potential biomarker is still not identified for BD. The authors also discussed the limitations of the study which helps the readers. Although the authors loosely commented on the future approaches, adding a more suggestive structured future direction will increase the impact of the manuscript.
Author Response
Thank you for the comments and suggestions. The future perspectives have been implemented as kindly suggested, to read: “Considering that BD disease is often described as an ensemble of phenotypes with different clinical characteristics, a future challenge could be to test if these phenotypes exhibit different miRNA patterns [61]. This could not only improve our knowledge on pathogenic processes underlying the various phenotypes, but also represent a step toward a more tailored therapeutic approach. To date, new “-omics “science, such as proteomics and metabolomics, could provide a comprehensive analysis of endogenous proteins and metabolites. In particular, with the use of metabolomics, one can potentially detect the alterations of physiological and pathological metabolites at the early stages of the disease due to its excellent sensitivity. In BD, two metabolomic tests have been developed and subsequently validated with reported high specificity and sensitivity [41][42]. Unfortunately, although this approach seems to be very promising, these tests are still expensive and not widely available, limiting to date their use in routine diagnostic or follow-up testing. In addition to metabolome investigations from blood samples, many studies have recently focused on analyzing the fecal metabolome alterations, resulting from changes in the gut microbial communities in BD patients [62][63]. Since the intercorrelation between diet and gut microbiota in well known, studying intestinal altered metabolic profiles and the microbial community imbalance of BD patients, is paving the way to new therapeutic approaches based on nutritional interventions [64].”
